# Generation Network for Echocardiographic Sectional Positioning and Shape Completion

## Abstract

The precise localization of 2D echocardiography planes in relation to a dynamic heart necessitates specialized expertise, as existing automated algorithms prmarily classify standard views while lacking the capability for comprehensive 3D structural perception. Traditional measurement techniques have evolved to infer 3D heart geometry, yet recent advancements in artificial intelligence, though demonstrating spatial awareness, still fall short in providing explicit 3D modeling. CTA-based digital twins, while promising, are hindered by cost and radiation concerns. Echocardiography, being cost-effective and radiation-free, remains limited in its ability to provide 3D perception. To address this gap, we introduce a novel point cloud-based weakly supervised 3D generation network specifically tailored for echocardiograms. This network automates 3D heart inference, and biomarker modeling, based on 2D echocardiography, slice tracking. To further enhance accuracy, we integrated a self-supervised learning branch into our framework, introducing multi-structure reconstruction loss and an overall reconstruction loss specifically designed for cardiac structure completion. Additionally, we constructed a comparative branch that serves to bolster the network's precision in inferring cardiac structures, thereby refining our approach and elevating the fidelity of the generated 3D models. Our approach enables real-time, robust 3D heart modeling, independent of paired data requirements, thereby facilitating research advancements in echocardiographic digital twins.

## 1 Introduction

In two-dimensional(2D) echocardiography, sensing the relative position between the sectional view and the dynamic heart is a professional task that only skilled ultrasound physicians can perform. In the past, many algorithms for automatically sensing the position of 2D ultrasound sections without magnetic positioning have been proposed, but most of them are aimed at static grayscale medical images and require the input of corresponding three-dimensional(3D) volume data at the same time during inference. (De Silva et al., 2013; Zhang et al., 2022; Wang et al., 2023) Although those registration based methods can locate the position of 2D ultrasound, their data requirements are high and it is difficult to make real-time inferences.In this regard, Luo et al. (2023) developed a freehand 3D reconstruction method for 2D fetal ultrasound scans based on online learning. While this method achieves ultrasound 3D reconstruction without magnetic localization in a practical sense, it still requires a significant amount of 3D ultrasound images during the model training process.

Unlike the linear scanning techniques frequently employed in ultrasound examinations of many other body parts,2D echocardiography often involves a sector scan within a fixed acoustic window to locate standard planes of the echocardiogram. Instead of linear scanning to detect potential lesions, echocardiography primarily focuses on the functional assessment of the heart's fixed structures. Consequently, determining the relative spatial relationship between the ultrasound slices and the heart is the initial step in scan guidance of echocardiography. Some efforts have been made to assess the relative relationship between ultrasound slices and the heart, yet these studies typically concentrate solely on the classification of standard views. (Madani et al., 2018; Wegner et al., 2022; Li et al., 2024b; Kusunose et al., 2020) Freitas et al. (2024) proposed a image-based method for plane localization in focused cardiac ultrasound. This study achieved the localization inference of ultrasound slices. However, their method necessitates a set of input images to determine the relative

3D poses between slices, lacking the capability to infer 3D structures and directly obtain the relative 3D pose between 3D heart structures and 2D ultrasound slices.

In reality, a pivotal objective in locating ultrasound planes is to perceive the 3D structure of the heart. It is crucial to tap into the spatial perception capabilities of neural networks in this regard, yet previous ultrasound AI models have failed to accomplish this precisely and directly. Essentially, many measurements in 2D echocardiography involve achieving 3D spatio-temporal perception of heart structure and function.(Nakajima & Shibutani, 2023) Taking the measurement of left ventricular ejection fraction as an example, from M-mode echocardiography to the Simpson's single-plane method, and then to the Simpson's biplane method, these measurement models are all based on the goal of inferring the 3D structure of the left ventricle, albeit proposed with different assumptions. (Otterstad et al., 2001; He et al., 2023)

In recent years, deep learning has seen rapid advancements. Ouyang et al. (2020) leveraging data-driven approaches, modeled left ventricular ejection fraction (LVEF) using video AI models. Specifically, they employed R2+1D to measure LVEF using a single plane, typically requiring two planes for measurement, demonstrating a form of spatial perception in AI models. However, they did not explicitly model the spatio-temporal information in echocardiography. Although 2D explicit modeling is often outperformed by data-driven regression methods (He et al., 2023), this does not negate the significance of explicit modeling of 3D structures. For instance, a study by Xu et al. (2023) illustrated that using a 3D label completion network can better model the fine structure of the left atrium, not only enhancing geometric assumptions for connecting structures such as the left atrial appendage and pulmonary veins but also surpassing the Simpson's biplane method in measurement performance, as outlined in current guidelines. Furthermore, explicitly modeling the 3D structure of the heart benefits computational biology (Camps et al., 2024; Li et al., 2024a), hemodynamic research (Karabelas et al., 2022), detailed observation of heart structures(Beetz et al., 2023), and analysis of cardiac dynamic patterns (Laumer et al., 2023). Such advancements provide a solid technical foundation for the development of digital twins of the heart.

Echocardiography currently represents the low-cost and non-radiative means of observing cardiac structures. In terms of dynamic observation of specific structures such as valves, 2D echocardiography holds advantages (Zoghbi et al., 2024). However, 2D echocardiography typically only captures motion within standard planes, limiting the ability to perceive the 3D structure of heart. The guidance of echocardiographic scanning has also remained an unresolved challenge. Although 3D ultrasound probes can be used to perceive the 3D structure of the heart, their resolution is inferior to that of 2D echocardiography. Furthermore, many cardiac interventional procedures rely on 2D transesophageal echocardiography for monitoring (Cutrone et al., 2024). The widespread use of 2D echocardiography makes it difficult to overlook its potential advantages and contributions in the process of creating digital twins of the heart. Therefore, there is a need to develop a spatial perception model based on echocardiography that can achieve explicit modeling of the heart's three-dimensional structure from 2D echocardiography images and rapidly locate the relative position of the current view within the heart's 3D structure.

Point cloud completion aims to address incomplete point cloud data resulting from occlusion, sparsity, or sensor limitations. Traditional point cloud completion methods(Yuan et al., 2018; Mao & Yang, 2023; Miao et al., 2024; Egiazarian et al., 2019; Huang et al., 2020) typically employs Multi-layer Perceptions (MLPs) to analyze each point separately before aggregating these insights into a comprehensive feature set via a symmetric operation, such as Max-Pooling. Some methods(Dai et al., 2017; Han et al., 2017) convert point cloud data into 3D voxel representation and then then processing it using 3D CNN. However, these methods need to increase voxel resolution to enhance the accuracy, which will lead to an explosion in computational costs. GRNet(Xie et al., 2020) and VE-PCN(Wang et al., 2021) adopt voxel grids as intermediate representation to tackle this problem.

With Transformer(Vaswani, 2017) being proposed for natural language processing tasks due to its excellent representation learning capabilities, recent efforts have increasingly focused on applying Transformer to point cloud completion to extract correlated features between points(Li et al., 2023; Yu et al., 2021; Wang et al., 2024; Yu et al., 2024).

In addition to these two mainstream approaches, other works have explored alternative strategies, such as(Lyu et al., 2021)leveraging the concept of diffusion models, (Wang et al., 2021) attempting to utilize the edge features of objects, and (Wu et al., 2021) proposing a new metric, DCD, inspired

by Chamfer Distance (CD) and Earth Mover's Distance (EMD). However, none of these works have considered the point cloud completion from entirely two-dimensional point clouds. Based on PCNs(Yuan et al., 2018), we have developed a novel weakly-supervised 3D generation network tailored for echocardiography, which is fully adapted for the task of completing point clouds from entirely two-dimensional point clouds.

The main contributions of this work are as follows:

- We innovatively propose a weakly-supervised single-view 3D generation network and processing pipeline based on point clouds for echocardiography. This system is designed to locate and track the displacement of echocardiographic views and enable single-view inference of dynamic heart structures.

- We constructed a lightweight neural network with multiple structural branches and local generative block(LGB) tailored for heart structure completion.

- We use contrastive reconstruction losses, enhancing the network's accuracy in inferring heart structures while simultaneously decoupling multi-structural point clouds of the heart. Our model achieved optimal performance on the test set.

Our method aims to fully automate, rapidly, and robustly complete 3D inference of heart structures. It does not rely on a large amount of paired training data and can perform real-time inference of heart 3D models and view localization during scanning.

## 2 METHOD

### 2.1 PROBLEM FORMULATION

Given a two-dimensional slice point cloud $X \in \mathbb{R}^{N_1 \times 2}$ of input, we need to find the full heart structure $Y_{Shape} \in \mathbb{R}^{N_2 \times 3}$ corresponding to the input, as well as the tangent plane of the input slice related to $Y_{Shape}$, denoted as $Y_{Rotate} \in \mathbb{R}^{N_1 \times 3}$, so that they match the actual corresponding $Y_{gtShape} \in \mathbb{R}^{N_2 \times 3}$ and $Y_{gtView} \in \mathbb{R}^{N_1 \times 3}$. That is to say, solving:

$$\min_{f, g_{view}, g_{shape}} L(Y_{gtShape}, g_{shape} \circ f(X)) + L(Y_{gtView}, g_{view} \circ f(X))$$

Among them, $f$ is the encoder. $g_{shape}$ and $g_{view}$ are the decoders. Since the loss function $L$ needs to measure the differences between point clouds, it is necessary to choose a loss function that is suitable for the type of point cloud data.

Based on cardiac shape point cloud $Y_{gtShape}$ get from segmentation of computed tomography angiography (CTA) and view point cloud $X$ get form 2D down sample of $Y_{gtShape}$, we utilize weak supervised learning to train multi-branch PCN $h(X) = (g_{view} \circ f, g_{shape} \circ f)(X)$ parameterized by weights $\theta$. When using, the input view point cloud $X$ can be get from echocardiographic segmentation mask.

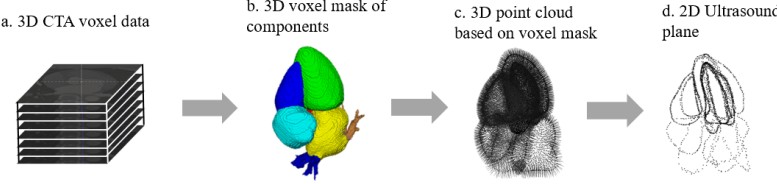

Figure 1: Training dataset processing pipeline. CTA imaging data obtained from XXX were segmented automatically using a previously described and validated 3D convolutional neural network. And we get normalize point clouds and echocardiographic planes (A2C, A4C, A5C and PLAX views) from voxel mask using open3d package (version 0.17.0).

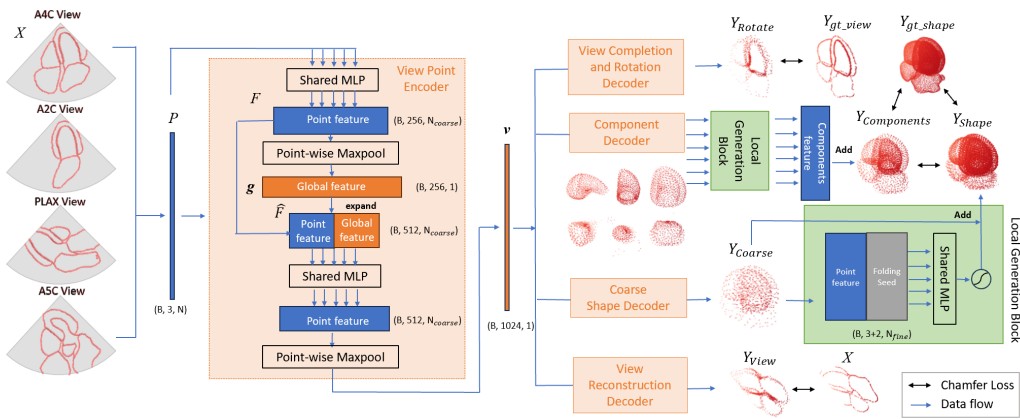

Figure 2: This image illustrates a multi-branch network structure for echocardiographic view localization and 3D structural inference. The network is based on PCN and incorporates self-supervised learning and multi-branch reconstruction loss. It uses an encoder for feature extraction and multiple parallel decoders for different tasks. The network has multiple structure branches to accurately depict each heart component, using component decoders and local generative blocks for structural reconstruction. Local generative blocks use folded seeds to model local structures without global feature integration. A view reconstruction branch enhances the network's understanding of input slices, improving slice localization and structural defect sensitivity for precise acoustic window and slice angle determination.

## 2.2 DATA PREPROCESSING

The dataset presented in this study was gathered using a GE Revolution Apex scanner over a one-year period, spanning from August 2023 to August 2024. The CTA images were automatically segmented by leveraging a previously validated 3D convolutional neural network (Xu et al., 2021), which provided ten labels, including the left atrium (LA), left ventricle (LV), right ventricle (RV), right atrium (RA), Left ventricular myocardium (MYO), aorta, and left/right pulmonary arteries. This method achieved a Dice score exceeding 95% when compared to manual segmentation, demonstrating its high accuracy. The quality of the 3D segmentation results was further visually inspected by senior radiologists to ensure they met our rigorous standards. Ultimately, a total of 2508 CTA voxel data were incorporated into this work.

To address the potential issue of data redundancy arising from the inclusion of multiple scans from the same patient at different time points, we split dataset in case-level to ensure that different samples from the same case do not inadvertently appear in both the training and testing sets. Specifically, we allocated a ratio of 7:1:2 for the training, validation, and test sets, respectively. This translates to 1007 cases (1778 scans) in the training set, 145 cases (255 scans) in the validation set, and 288 cases (475 scans) in the test set.

The process of obtaining heart structure and slice point clouds for training and testing purposes involves intricate steps, as shown in Figure 1. The heart structure point cloud requires edge coordinate sampling, normal generation, point cloud resampling, pose calibration, and standardization of point cloud size. On the other hand, generating slice point clouds entails section positioning, acquisition of section point clouds, generation of section mask images, image-based 2D point cloud acquisition, and simulation of visible structural incompleteness within the sections. Detailed descriptions and code of these procedures are provided in the supplementary materials.

## 2.3 WEAKLY SUPERVISED MULTI BRANCH PCN

### 2.3.1 MULTI BRANCH NETWORK STRUCTURE

We constructed a multi branch completion network incorporating self-supervised learning and multi-branch reconstruction loss based on PCN (Yuan et al., 2018).We adopted the encoder structure of

PCN as the feature extractor and paralleled multiple decoders to achieve echocardiographic view localization and 3D structural inference tasks. The detailed workflow is illustrated in Figure 2.

**Multi Structure Branches:** In this study, the full-heart point cloud consisted of six components, including the LV, LA, MYO, RV, RA, and aorta. When considering individual structures, the MYO exhibited significant concave features. When considering the global structure, coupling occurred among structures at valve rings. To accurately depict each component's structure, we needed to perform end-to-end structural decoupling. This decoupling was necessary for the full-heart point cloud generated by the network, to prepare it for subsequent testing. Therefore, we employed component decoders and local generative blocks for structural reconstruction. The component decoder generated point clouds for the six heart structures, while the coarse shape decoder produced the full-heart shape as one point cloud directly. After passing through the local generative blocks, the full-heart structural point cloud could be characterized from two perspectives.

**Local Generative Blocks:** Directly using the component decoder and coarse shape decoder for mutual supervision of the two generated full-heart point clouds would interfere with network optimization under supervision by real point cloud data. To address this issue, we employed folded seeds to model the local structure of the point cloud. However, in the original setting of PCN, local structure modeling should extend global features and merge them with point features and folded seed features to form features with a width of 1029 for processing. Considering that the completion network in this study is dedicated to the heart and does not face the challenging implicit classification problem of various fine-grained target point clouds as in ShapeNet, the local generative blocks in this study are not connected with global features $v$. Additionally, we noted that merely using MLPs made it difficult to depict curved morphologies locally. Therefore, we incorporated a Sigmoid setting as follows:

$$Y_{Shape} = \lambda(2Sigmoid(MLP(\hat{Y}_{coarse}, seed)) - 1) + \hat{Y}_{coarse}$$

where $\hat{Y}_{coarse} \in \mathbb{R}^{16384 \times 3}$ is the result of $Y_{coarse} \in \mathbb{R}^{1024 \times 3}$ after broadcasting, $\lambda$ is a hyperparameter with a value of 0.1.

**View Reconstruction Branch:** We utilized a view reconstruction branch to enhance the network's understanding of the input slices. Since our input slices were cropped according to the echocardiographic sector and view window, the branch used for slice localization in this network actually served both slice localization and slice structure completion functions. Moreover, the cropping pattern of the slices is related to the acoustic window and contains slice information. Therefore, we designed a slice reconstruction brunch to enhance the stability of slice shapes and make the network more sensitive to structural defects related to slice angles, enabling more precise localization of the acoustic window and slice angles.

### 2.3.2 Contrastive reconstruction losses

Due to the use of contrastive reconstruction strategy in this network, the loss function is divided into three parts:

The first part is $L_{coarse}$, which includes sparse point cloud chamfer loss for six output structures, chamfer loss for reconstructed and rotated completed cross-sections, and loss for overall shape branching, as this network focuses on both whole heart structure and cross-section localization.

The second part is $L_{fine}$, which further supervises the shape formed by merging six output structures and the overall shape output by the shape branch after using the branch of the local generation block.

The third part is the contrastive reconstruction loss $L_{compare}$, which supervises the differences in the overall shape output by merging six output structures and shape branches. Enable the two branches to better promote each other. The total loss is as follows:

$$L_{total} = L_{coarse} + \alpha L_{fine} + \alpha\beta L_{compare}$$

where $\alpha$ and $\beta$ are hyperparameters, and $\beta$ is always 0.1, while $\alpha$ increases with training and becomes 0.01 when the epoch is less than 100 and 0.1 when the epoch is more than 200. Namely, in the early stages of training, the loss function makes the network pay more attention to the overall shape of the rough point cloud. In the later stages of training, the loss function makes the network pay more attention to the characterization of fine shapes and the similarity between the two branches. Due to the multiple output branches of the network, we need to consider the loss of each supervision

separately. All the above losses are based on the L1-norm based chamfer distance, which reads:

$$L_{CD}(S_{output}, S_{gt}) = \frac{1}{S_{output}} \underset{x \in S_{output}}{\Sigma} \underset{y \in S_{gt}}{\min} ||x - y||_{L1} + \frac{1}{S_{gt}} \underset{y \in S_{gt}}{\Sigma} \underset{x \in S_{output}}{\min} ||x - y||_{L1}$$

where $S_{output}$ is output point cloud, and $S_{gt}$ is ground truth. So the loss functions are as follows

$$L_{coarse} = \underset{x \in \Omega}{\Sigma} L_{CD}(x, x_{gt}) + L_{CD}(Y_{coarse}, Y_{gtShape}) + L_{CD}(Y_{View}, X) + L_{CD}(Y_{Rotate}, Y_{gtView})$$

$$L_{fine} = L_{CD}(Y_{Shape}, Y_{gtShape}) + L_{CD}(Y_{Component}, Y_{gtShape})$$

$$L_{compare} = L_{CD}(Y_{Component}, Y_{Shape})$$

where $\Omega$ is coarse component set $\{S_{lv}, S_{la}, S_{rv}, S_{ra}, S_{myo}, S_{aorta}\}$, and $x_{gt}$ means respective ground truth point cloud of $x$.

## 3 RESULTS AND ANALYSIS

### 3.1 IMPLEMENTATION DETAILS

The models were implemented using PyTorch on an NVIDIA GeForce RTX 3080 GPU. A batch size of 16 and Adam optimiser with a learning rate of 10-4 were used for all models. We trained each model for 200 epochs using a 0·7 decay learning rate scheduler with 50 epochs. The performance on validation set was recorded to optimise the network and set up the hyperparameters. We used various metrics to assess the model's performance on shape reconstruction and view localization tasks, including L1-norm based chamfer distance(CD), L2-norm based chamfer distance, FScore with threshold as 2mm. Other than that, we use degree error, and center distance error of view planes to assess view localization function of our model.

### 3.2 SHAPE RECONSTRUCTION

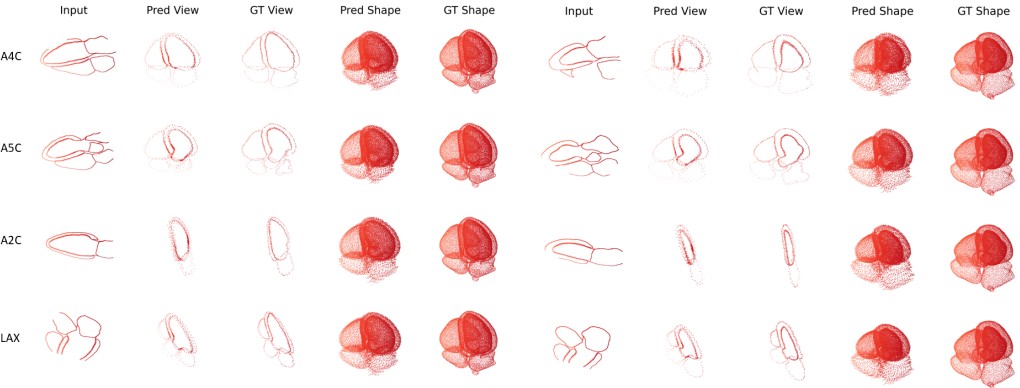

Figure 3: The visualization of the results of sectional localization and shape completion for two cases from different views.

The use of local generative block can improve the generative performance of PCN. And the component brunch made it possible to get heart structure separately from network output.When the sparse output of component branches is not finely characterized using local generation blocks, directly using contrastive reconstruction loss will lower the performance of the overall shape point cloud output. When using locally generated blocks, the output accuracy of model component branches is significantly improved and can exceed that of the overall shape point cloud, as shown in Table 1.

| Model | Shape Brunch Output | | | Component Brunch Output | | |
|---|---|---|---|---|---|---|
| | L1 CD(mm) | L2 CD(mm) | FScore(2mm,%) | L1 CD(mm) | L2 CD(mm) | FScore(2mm,%) |
| PoinTr(Yu et al., 2021) | 1.978 | 0.0644 | 61.43 | - | - | - |
| PCN(Yuan et al., 2018) | 2.128 | 0.0694 | 59.76 | - | - | - |
| PCN+SLGB | **1.874** | **0.0534** | 65.33 | - | - | - |
| PCN+Com+SLGB | 1.930 | 0.0572 | **65.58** | 2.236 | 0.0726 | 50.09 |
| PCN+Com+SLGB+Rec | 1.940 | 0.0576 | 65.16 | 2.250 | 0.0738 | 49.63 |
| Ours(PCN+Com+SLGB+CLGB+Rec) | 1.938 | 0.0578 | 65.22 | **1.550** | **0.0372** | **79.55** |

Table 1: The performance of models.Com represents whether to use component branches, Rec represents whether to use contrastive reconstruction loss, SLGB represents whether to use the local generative blocks mentioned above in the coarse shape output branches, and CLGB represents whether to use local generative blocks in the component branches

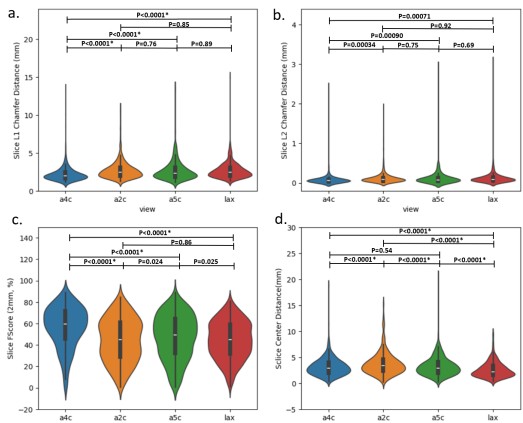

Figure 4: The performance distribution of the network in cross-sectional localization using a violin plot. Among them, a) and b) respectively show the L1 and L2 chamfer distances between the output slice and the actual slice. c) Displayed FScore with a threshold of 2mm on different cross-sections. d) The distance deviation of the center of the plane after fitting the plane with the cross-section is shown. Overall, the network has good segmentation ability in different aspects.

Our network can accurately restore shapes from a single view, as shown in Figure 3. Our method not only effectively decouples the multi-chamber shapes of the whole heart in terms of shape reconstruction but also achieves optimal performance in shape reconstruction accuracy. The F-Score presented in the table represents points that lie within a distance of 0.01 from the ground truth point cloud. Based on the dimensional information of CTA and our data processing pipeline, distance of 0.01 in our point cloud corresponds to a real-space error of 2 millimeters, implying that nearly 80% of the points in the shapes generated from a single slice are within 2 millimeters of the ground truth point cloud. Achieving an error within 2 millimeters is a leading goal in current cardiac surgery guidance technology, which demonstrates that our network can effectively infer and reconstruct the whole heart structure from a single slice. For detailed reconstruction performance across various slices and numerical values of reconstruction performance for structures visible or invisible in the current slice, please refer to Figure 5(a) and (b). Currently, among the overall dataset, the A4C view exhibits the best reconstruction performance. And the generation result of A2C view is the worst. This may be attributed to the fact that the A4C view directly reflects the structural information of the four chambers and the left ventricular myocardium, with only one invisible structure, the aorta. In contrast, the A2C view only reflects partial information of the LV, LA, and left ventricular myocardium, introducing some uncertainty about invisible parts and resulting in slightly poorer reconstruction performance. However, overall, there are no significant differences in reconstruction performance among these slices.

| | A4C | A2C | A5C | PLAX | All |
|---|---|---|---|---|---|
| $\theta_{3D}(°)$ | $2.79 \pm 4.85$ | $5.75 \pm 4.13$ | $4.98 \pm 4.22$ | $5.79 \pm 3.89$ | $4.83 \pm 4.45$ |
| $d_{3D}(mm)$ | $3.17 \pm 1.73$ | $3.78 \pm 2.22$ | $3.24 \pm 1.82$ | $2.61 \pm 1.51$ | $3.20 \pm 1.88$ |

Table 2: The performance distribution of networks in view localization. Among them, $\theta_{3D}(°)$ is the normal angle deviation between the three-dimensional plane fitted by the inferred section and the corresponding three-dimensional plane of the real section, measured in degrees. $d_{3D}(mm)$ is the distance between the centroids of the predicted and ground truth 3D view plane. To calculate the centroid position, the boundary of two 3D planes is defined as the boundary with the inferred epicardium.

### 3.3 VIEW POSITION

Our method performs stably and accurately in slice localization. The L1-norm and L2-norm based chamfer distances between the output slice and the ground slice are relatively stable, but there are differences in FScore, as shown in Figure 4. Overall, the effect of the A4C view is the best, and the positioning effect of the PLAX and the A2C view is slightly lower than that of the A4C view. The overall angle error is 4.83 ± 4.45 degrees, with a four chamber heart rate of 2.79 ± 4.85 degrees, as shown in Table 2. In fact, the focus of A2C view is to observe the mitral valve, and it is difficult to ensure that the section with the largest inner diameter of LV is obtained. In cases where the LV is relatively symmetrical, the eligible A2C view may have two corresponding symmetrical slices, leading to an increase in error.

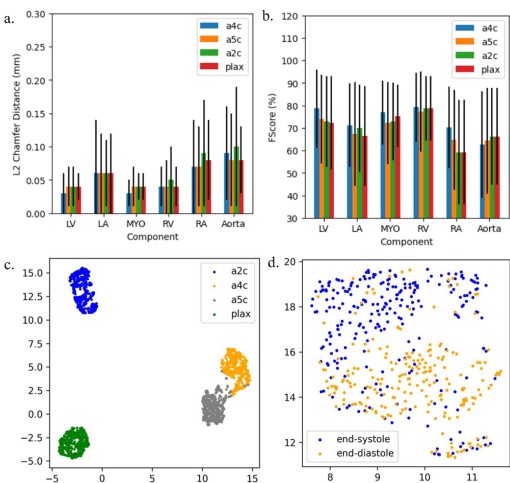

Figure 5: a) and b) show the performance of different related structures on different sections, where dark blue dots are used to color the heart structures that are not visible on the current section, and yellow dots are used to color the heart structures that are visible on the current section. Different line colors represent different input facets. c) The dimensionality reduction distribution of the hidden layer encoding of input slice samples by the network is demonstrated, where the hidden layer encoding is the v vector in the text, indicating that the network correctly identifies the type of slice. d) The distribution trend of feature vectors encoded by different slice networks in the same case in terms of time is shown. The dimensionality reduction method used in c) and d) is umap.

### 3.4 REPRESENTATION LEARNING

In terms of structural inference, we aimed to avoid catastrophic mode collapse in the network's predictions of structures not visible in the current single slice, which fortunately did not occur. In fact, as shown in Figure 5 (a), (b), for structures invisible on the input view, the network's performance was only slightly inferior to that for visible structures. Regarding the accuracy of structural inference,

the complexity of the structure itself had a greater impact on the network's predictions. Indeed, for the aorta, there may have been deviations in boundary definitions within this batch of training data, as voxel segmentation was limited by the coverage of CTA, while point cloud completion inference might notice longer or shorter aortic structures.

Simultaneously, the network demonstrated a certain level of representation learning ability, as shown in Figure 5(c),(d). We used UMAP package (version 0.5.6) to reduce the dimensionality of the feature vectors output by the encoder of network and found that it exhibited excellent slice classification performance when encoding individual slice samples. Furthermore, it captured the similarity between slices in their adjacent relationships. For example, a A5C view, sometimes similar to a A4C view in terms of marginal structures when the aorta is not clearly visible. When encoding multiple slices of each heart and performing combined dimensionality reduction analysis, the network was able to reflect temporal trends. Even though the our network did not consider the temporal correlation of the dynamic heart and trained the end-systolic and end-diastolic phases as separate structural data, there was still some clustering effect in the feature space between these two phases, despite the network not receiving any arbitrary input or supervision regarding cardiac phases.

The heart is a dynamic system in which various structures interact with each other, and some local structural features may affect the overall motion and health of the heart. These observations reflect that the proposed network can be a representation learning model, may encode representations that imply more information about the heart, even though it was not directly optimized by relative supervisory signals.

## 4    CONCLUSION

We have made a groundbreaking contribution by introducing a novel point cloud-based, weakly supervised, single-view 3D echocardiography generation network, along with a comprehensive processing pipeline tailored to it. This methodology not only achieves precise localization and tracking of echocardiographic slice displacements but also successfully applies to the inference of cardiac dynamic structures from a single plane. By converting voxel data into point clouds and employing a series of innovative processing tools, we have constructed an efficient and lightweight neural network. Furthermore, the introduction of multi-structural reconstruction loss, local generative blocks, and contrastive loss has significantly enhanced the accuracy and robustness of cardiac structure inference. Most importantly, this method aims to achieve fully automated, rapid, and highly robust 3D inference of cardiac structures. It is capable of generating real-time 3D heart models and performing slice localization during scanning, providing potent support for clinical decision-making.

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

## A   DATA PREPROCESSING AND DISTRIBUTION

### A.1   VOXEL DATA TO POINT CLOUD DATA

For the cardiac data processing, we first read the voxel data of the three-dimensional heart structure and convert it into a 3D array. Then apply an affine transformation to align it with a unified coordinate system. For the annotated cardiac chamber structures and vascular structures within the voxel data, we iteratively traverse these structures, treating each one as a positive sample and examining all its eight-neighborhood voxels. By identifying the coordinates of voxels with negative labels, we determine the boundary point clouds for each structure, thus accurately extracting the inter-structural boundary information. Different structures are stored with distinct color information for differentiation. Specifically, valve points are located at the intersection of chamber annotations, with the mitral valve taking the intersection between the left ventricle and left atrium, the tricuspid valve at the intersection between the right ventricle and right atrium, and the aortic valve at the intersection between the left ventricle and aorta. Based on the voxel annotations, we sampled the following

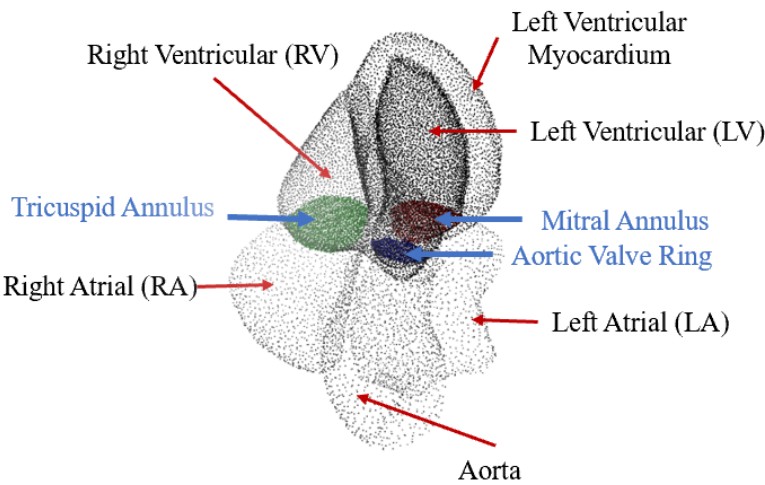

Figure 6: The schematic diagram of the structure corresponding to the point cloud, where the chambers and aorta are colored with varying degrees of gray, and the valve annulus are colored. Among them, green represents the tricuspid valve annulus, red represents the mitral valve annulus, and blue represents the aortic valve annulus.

structures: left ventricle, left ventricular wall, right ventricle, right atrium, left atrium, aorta, mitral valve, tricuspid valve, and aortic valve. These structures are illustrated in Figure 6.

To enhance the quality of the point cloud data, we also introduced a normal generation and refinement step. Specifically, we first identify the inner and outer points relative to the current structure of interest by scanning the eight-connected neighborhood of each voxel. The direction from the centroid of the inner point cloud to the centroid of the outer point cloud is defined as the initial simulated normal vector. We then smooth the simulated normal vectors using a minimum spanning tree algorithm to obtain more accurate normal vector information. For a voxel coordinate $(i, j, k) \in V$, if it is a boundary point, we consider whether the 26 neighboring voxels within the cube $[(i - 1) : (i + 1), (j - 1) : (j + 1), (k - 1) : (k + 1)]$ belong to the current structure or not. The normal vector at this vertex is determined by the direction from the centroid of the inner point set to the centroid of the outer point set. The generation of boundary points can be performed in parallel with the normal generation. Next, we use the inner and outer point clouds, along with their respective centroids and the initial simulated normal vectors, to grid the voxel data. We then use curvature-based point cloud resampling to standardize the number of points for the cardiac chamber and vascular structures within the gridded data, resulting in voxel-reconstructed point cloud data. Additionally, we standardize the number of points for each structure: 4038 points for the left ventricle, 4660 points for the left ventricular wall, 1299 points for the left atrium, 1000 points for the mitral valve, 3814 points for the right ventricle, 1480 points for the right atrium, 1000 points for the tricuspid valve, 1139 points for the aorta, and 1000 points for the aortic valve.

## A.2 POINT CLOUD DATA REFINEMENT

After obtaining the point cloud data, we perform point cloud pose correction and size normalization to acquire standardized whole-heart point cloud data. Point cloud pose correction involves defining and unifying the spatial coordinate system, while size normalization aims to enhance the consistency of the mean point coordinates across different sample point clouds, thereby reducing the difficulty of fitting the echocardiographic scan-guided network. Specifically, we use the centroid of the whole heart as the origin of the spatial coordinate system, the direction from the tricuspid valve centroid to the mitral valve centroid as the X-axis, the vector from the left ventricle centroid to the apex as the Y-axis, and the cross product of the X-axis and Y-axis direction vectors as the Z-axis. The corresponding affine matrix is automatically extracted based on the structural annotations. After

pose calibration, the vertex coordinates of the point cloud data are scaled down to $\frac{1}{300}$ of their original values, resulting in standardized whole-heart point cloud data.

### A.3 STANDARD VIEW ACQUISITION

Finally, based on the standard views specified in two-dimensional medical imaging standards for the heart, we segment and collect the whole-heart point cloud data to obtain standard view point cloud data. Specifically, we solve for the plane equations using three non-collinear key points for each standard view and traverse the standardized point cloud data, sampling the whole-heart point cloud data using the inequalities of the plane equations. The standardized view point clouds include the apical five-chamber view, apical four-chamber view, apical two-chamber view, and parasternal long-axis view. These views are illustrated in Figure 7.

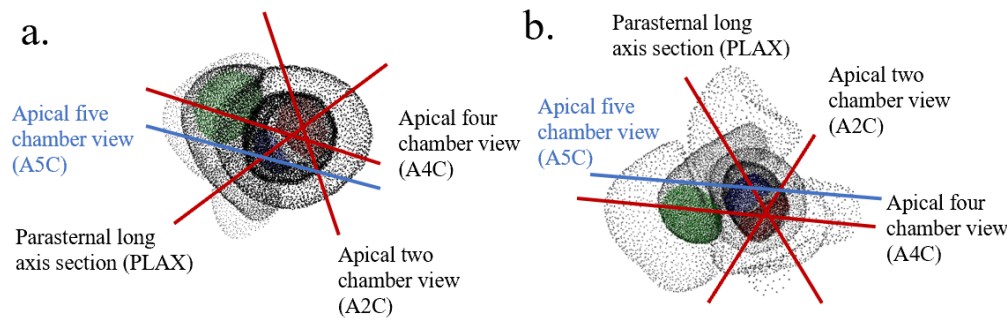

Figure 7: Definition of the A2C, A4C, A5C and PLAX views. The chambers and aorta are colored with varying degrees of gray. And green represents the tricuspid valve annulus, red represents the mitral valve annulus, and blue represents the aortic valve annulus. The solid red and blue lines indicate the A2C, A4C, PLAX, and A5C views. a) is the perspective from the apex to the base of the heart, and b) is the perspective from the base to the apex of the heart.

### A.4 DATA DISTRIBUTION

We conducted an analysis of the data utilized in this study. For each data, we established the spatial coordinate system with the centroid of the whole heart as the origin. The direction from the tricuspid valve centroid to the mitral valve centroid was defined as the X-axis, the vector from the left ventricle centroid to the apex as the Y-axis, and the cross product of the X-axis and Y-axis direction vectors as the Z-axis. Consequently, for the estimation of heart size distribution, it suffices to collect statistics along the X, Y, and Z axes.

Statistical analysis revealed that, for the training set, the mean minimum value on the X-axis is $-58.67mm$ with a standard deviation of $5.94mm$, and the mean maximum value is $50.86mm$ with a standard deviation of $5.08mm$. The mean minimum value on the Y-axis is $-80.84mm$ with a standard deviation of $7.61mm$, and the mean maximum value is $64.98mm$ with a standard deviation of $5.96mm$. The mean minimum value on the Z-axis is $-43.02mm$ with a standard deviation of $4.75mm$, and the mean maximum value is $54.68mm$ with a standard deviation of $6.17mm$. For the test set, the mean minimum value on the X-axis is $-59.58mm$ with a standard deviation of $6.49mm$, and the mean maximum value is $51.23mm$ with a standard deviation of $5.27mm$. The mean minimum value on the Y-axis is $-81.59mm$ with a standard deviation of $8.23mm$, and the mean maximum value is $64.92mm$ with a standard deviation of $6.12mm$. The mean minimum value on the Z-axis is $-43.59mm$ with a standard deviation of $5.09mm$, and the mean maximum value is $55.15mm$ with a standard deviation of $6.25mm$.

The aforementioned data indicate that the distributions of the training and test sets are consistent. A more detailed visualization of the width statistics for each axis of the heart is provided in the following figure.

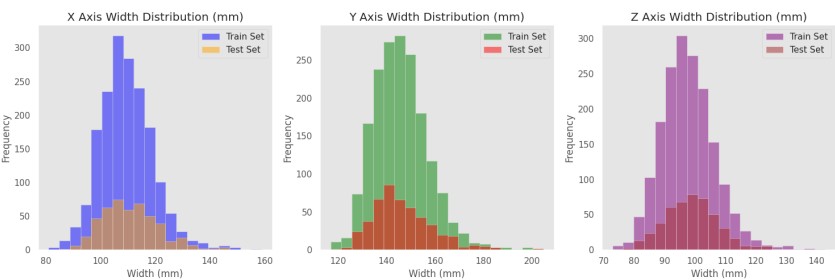

Figure 8: The histogram of the width distribution along each axis.

## B  HOW CAN THIS MODEL BE APPLIED TO CARDIAC ULTRASOUND?

### B.1  PAIRED DATA TESTING OF ULTRASOUND AND CTA

This model is used for generating three-dimensional structures from two-dimensional echocardiography, with the aim of inferring three-dimensional structures from two dimensions and performing sectional localization and guidance of two-dimensional echocardiography. The way to complete this function is not only based on contour point cloud completion. Using grayscale direct conversion is one method, but in previous experience, such methods were often limited by the data quality of simulated data and prone to mode collapse during real data testing. Therefore, some people have trained non paired data based on the CycleGAN approach, using latent spatial transformation to complete the conversion between ultrasound videos and cardiac grids. However, the performance of the above methods is limited and comprehensive biomarker performance testing has not been conducted.

We believe that there are several aspects of error between cardiac ultrasound and three-dimensional shape conversion. Firstly, the low signal-to-noise ratio of ultrasound makes it difficult for the model to accurately capture the shape edge patterns contained in ultrasound images. Secondly, the accuracy of converting two-dimensional shape edge patterns into three-dimensional spatial structures is limited. Using a network architecture to overcome two sources of error may not be accurate enough. The first source of error we will propose in another study is the Ultrasound Cardiac Whole Structure Segmentation Network (unpublished). This article aims to test the error situation of the two-dimensional to three-dimensional process and try to solve the second source of error as much as possible.

The testing process is divided into several aspects. On the one hand, the output should accurately reflect the properties of the ultrasound. On the other hand, the output should match the true three-dimensional shape of the patient's heart, that is, consistent with the structural information reflected by CTA. Therefore, we additionally collected parasternal long axis section ultrasound and CTA (from different hospitals) from 11 patients, calculated the left ventricular width at the end of ultrasound systole (US LVIDs, us2dlv), left ventricular width at the end of CTA systole 4-chamber view (CTA LVIDs, cta2dlv), left ventricular volume at the end of CTA systole (gt3dlv), and the correlation between the three-dimensional left ventricular volume (pred3dlv) completed by the ultrasound segmentation model in this article. The ultrasound segmentation model mentioned here is temporarily based on MTANet. It can be seen that in the three-dimensional structure inferred by this method, the Pearson correlation between left ventricular volume and the corresponding CTA's true left ventricular volume is high (0.69, p=0.018), and the Pearson correlation between right ventricular volume and the corresponding CTA's true right ventricular volume is high (0.82, p=0.0021).

### B.2  ULTRASOUND VIDEO PREDICTION OF 3D BIOMARKERS

In addition to accurately predicting three-dimensional structures using 2D ultrasound, another major clinical application of this model is to obtain non radiative and high-resolution 3D biomarkers. For CTA, special gating settings are required to control the radiation dose from end systolic to end diastolic to achieve multi frame 3D data acquisition, and usually only end systolic and end diastolic phases are collected or used for coronary observation. Collecting more than 10 CTA body data for one cardiac cycle is not a cost-effective and common practice. In contrast, two-dimensional echocar-

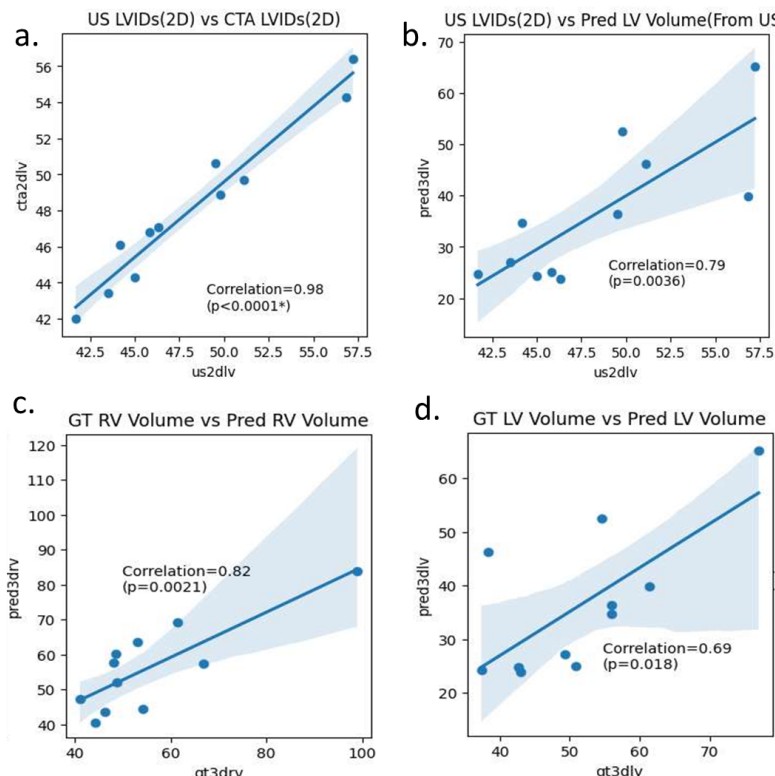

Figure 9: The correlation between ultrasound end systolic left ventricular width (US LVIDs, us2dlv), CTA end systolic 4-chamber view left ventricular width (CTA LVIDs, cta2dlv), CTA end systolic left ventricular volume (gt3dlv), and 3D left ventricular volume (pred3dlv) completed using the model in this article after ultrasound segmentation

diography has higher temporal resolution, sufficient spatial resolution, and no radiation. This work is directly based on the contour data of two-dimensional echocardiography to obtain cardiac structures, which may greatly increase the reserve of cardiac structural data, help expand the available data and refine the temporal resolution for structural heart disease research, and accelerate the research process of structural heart disease. Here is an example of frame by frame synchronous inference of an ultrasound video. The curve below the image shows the change in left ventricular volume.

The segmentation and structural inference effects of five time points in a cardiac cycle are shown above, and the corresponding time points are marked in the curve graph. The segmentation result is based on MTANet. It can be seen that the three-dimensional structure of the heart derived is basically synchronized with the changes in left ventricular volume and two-dimensional ultrasound.

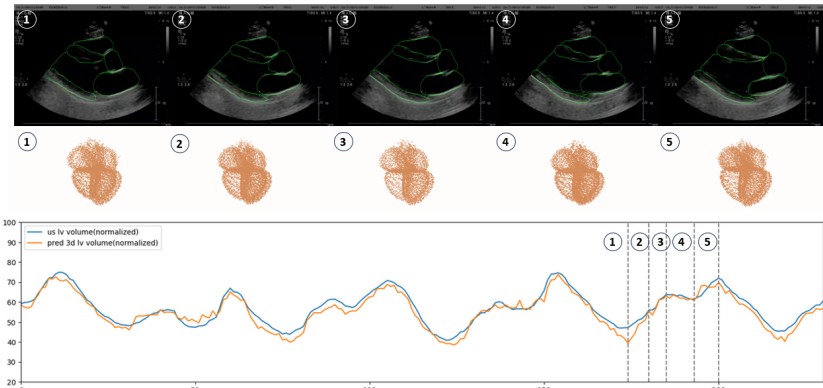

Figure 10: The change curve of left ventricular volume and the segmentation and structural inference effect at five time points during a diastolic process.

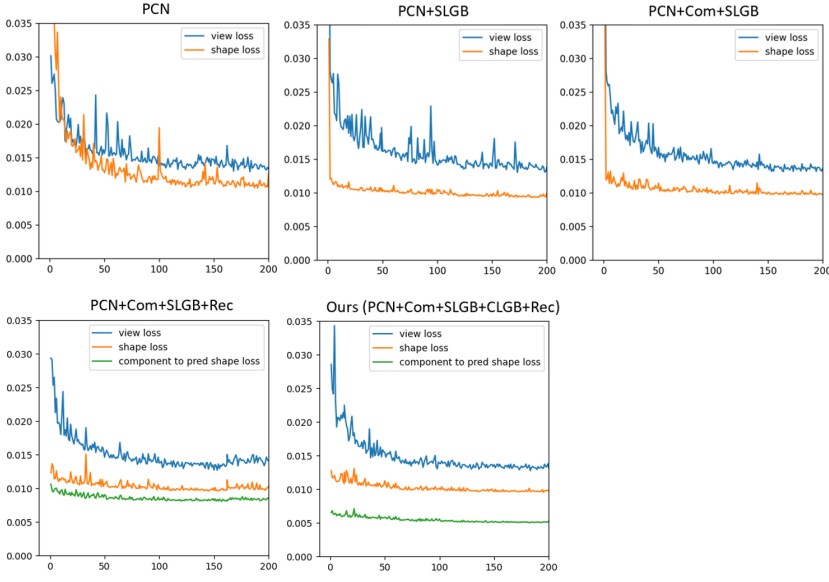

Figure 11: The trend of loss value change on the validation set within 200 epochs during models training.