# OpenReview forum: "Generation Network for Echocardiographic Sectional Positioning and Shape Completion"
_ICLR.cc/2025/Conference — ICLR 2025 Conference Withdrawn Submission_

### Official Review · Reviewer_t8bS · 2024-10-28

**Soundness:** 3
**Presentation:** 2
**Contribution:** 3
**Rating:** 5
**Confidence:** 3

**Summary:**

This paper first analyzes the current state of 3D heart modelling and then proposes a method for 3D heart reconstruction based on single 2D echocardiography plane. This paper sets up its learning task by the following steps:
- Introducing a data-preprocessing pipeline, by converting the existing 3D CTA voxel data into 3D point-cloud data and simulate the echocardiographic planes
-  Decoupling the reconstruction task into the coarse-shape, component and view reconstruction in a PCN for better learning objective.
-  Introducing contrastive losses between coarse-shape and component reconstruction branch.

**Strengths:**

This work studies an interest problem that might have huge potential clinical impact.

The authors well delivers the motivation of the paper.

The authors demonstrate a novel approach to solving the reconstruction problem.

**Weaknesses:**

This paper lacks a summary of the recent relevant research. For example, the authors should include some paragraphs to summarize the point-cloud reconstruction works other than PCN.

The methodology and experiment parts are generally not well-written. For example, in Figure 1, what does XXX mean? Figure 2 lacks of well explanation of each components. In multi-branch network structure part (2.3.1), the paragraph multi structure branches should be introduced first.

The training data-preprocessing pipeline is heavily rely on the segmentation network. However, the author does not explain or discuss how the proposed mothed comparison between other data-preprocessing methods.

The authors claims that the network is light-weight, but in the experiment part, it is hard to see how this statement is hold, since there is no analysis or comparison to show the statement.

The works lacks comparison between other existing 2D-to-3D cloud-point reconstruction methods.

**Questions:**

Adding more background information and literature summarization on 2D-to-3D point cloud rescontruction to the paper.

Adding comparison with modern network architectures, such as Transformer-based method or adding analysis to demonstrate why the method is lightweight.

Adding comparison between other data-preprocessing, for example, add more views, since there is no upper bound in this work (a fully supervised method).

In section 2.3.2, why there is contrastive loss term for input $x$ and $x_{gt}$ and what is the difference between notation $X$ and $x$?

---

### Official Review · Reviewer_gCnM · 2024-10-28

**Soundness:** 3
**Presentation:** 2
**Contribution:** 2
**Rating:** 3
**Confidence:** 5

**Summary:**

This paper proposes a framework for predicting 3D cardiac structural perception from 2D echocardiography planes. This is a new task in the echocardiographical domain that aims to explore the 3D cardiac structures instead of using CTA. All experiments are conducted in their in-house dataset.

**Strengths:**

1. This paper proposes a new task in echocardiography, which has enabled 3D cardiac structural prediction via using 2D echocardiographic images.

2. This paper demonstrates the good performance in predicting cardiac structures.

**Weaknesses:**

1. First, this paper can benefit from releasing the dataset since all the designs serve this dataset, which has a large amount of CTA data corresponding to cardiac structures. Such a dataset can help train a robust network for accurate cardiac structural prediction.

2. A drawback of this task is using only the segmentation (cardiac structure contours) predicted by the 3D CNN network from CTA images for 3D shape prediction. We are not able to ensure the echocardiography scanned in the real scenarios can also make the closed shape compared with the prediction from CTA. For example, the CTA scans always have a fixed position for cardiac, and the imaging quality is much better than the echocardiography. In contrast, due to the different imaging principles, echocardiography often has relatively large distortion with poor image quality, and image acquisition highly depends on the sonographer's experience.

3. With weakness 2, I consider that the experiment can be improved by using some image pair. The author can collect both CTA and echocardiography data from the same patient/person. Then, the experiment can utilize real echocardiography to predict the 3D cardiac structures. With this experiment, the author can also demonstrate that the proposed approach has overcome the following points:
i). What is the domain gap between CTA and echocardiography when applying this method? ii). Can this method actually be applied in echocardiography? iii). With the pair of CTA and echocardiography, the result could be more convincing.

4. The network is designed only for this task; I don’t think this network has much innovation because all modules and designs are integrated with other methods. For example, global and local features, coarse to fine, some argumentations, etc.

**Questions:**

1. Will the author release the dataset? I think this work is 80% dependent on their dataset and thier task.

2. Can an additional experiment be added to the rebuttal? It is really important to validate that these newly proposed tasks can benefit real medical applications and inspire follow-up works.

**Details Of Ethics Concerns:**

If the author releases the dataset, then an ethics review is required. However, this paper does not demonstrate whether this dataset will be publicly available or not.

---

> ### Comment · Reviewer_gCnM · 2024-12-02
> **Respond to Rebuttal**
>
> Response 2: The question is that echocardiography often has relatively large distortion with poor image quality, as the author said that 'CTA and ultrasound models have differing boundary definitions, leading to size over/underestimations'. Directly using the segmentation mask is easily affected by the domain gap in real scenarios (The author should address this problem, as I mentioned in weakness 2). More experiments are needed to verify the generalizability of this method.
>
> Response 3 & 6: Since Figure 10 is the experiment? The author should point out where you add extra experiments in this paper first. Also, Figure 10 only shows that the normalized area of CTA and echocardiography have a similar change in cardiac structures during the heartbeat cycle, which can not illustrate that predicted 3D cardiac structures from echocardiography align with the real CTA (The chamber size, shape and morphology).
>
> Response 1 & 5: No comments
>
> Response 4: No comments
>
> I would keep my rating because I think my questions have not been addressed yet, and the author should point out where they add experiments or modifications in the paper.

---

### Official Review · Reviewer_WVb4 · 2024-11-03

**Soundness:** 1
**Presentation:** 1
**Contribution:** 2
**Rating:** 1
**Confidence:** 4

**Summary:**

The paper presents a weakly-supervised 3D generation network for echocardiographic sectional positioning and shape completion. The paper aims to address the limitations of traditional methods in providing explicit 3D modeling of heart structures from 2D echocardiography images. The proposed network uses point clouds to infer cardiac structures and can perform real-time inference without requiring significant paired training data. The authors also integrate a self-supervised learning branch into their framework, which enables multi-structure reconstruction loss and overall reconstruction loss for cardiac structure completion. Experimental results demonstrate superior performance on the test set, showcasing the potential of this approach in facilitating the reconstruction of heart digital twins form echocardiography.

**Strengths:**

- The paper addresses a significant problem in the field of echocardiography, namely the lack of efficient and accurate methods for inferring 3D heart structures from 2D ultrasound images. Current methods rely on manual segmentation or registration-based approaches that are time-consuming, labor-intensive, and often require extensive expertise.
- The proposed approach requires only echocardiography as the input modality, which is a widely used, low-cost, and non-radiative technique in clinical practice. This is a significant advantage because it eliminates the need for additional imaging modalities, making it more practical and accessible to a wider range of healthcare settings.
- The proposed weakly-supervised single-view 3D generation network and processing pipeline based on point clouds for echocardiography can address the limitations of traditional methods.
- The proposed network leverages the spatial perception capabilities of neural networks to infer 3D structures and directly obtain the relative 3D pose between 3D heart structures and 2D ultrasound slices.
- The approach enables real-time inference making it suitable for applications that require rapid responses.

**Weaknesses:**

- The paper has a poorly written methods and results sections, with incomplete and ungrammatically structured sentences (e.g., 131-135). This makes it difficult for readers to understand the methodology and results.
- The presentation of the network's components is not logical or well-articulated, making it challenging to follow the authors' reasoning and design choices.
- The supplemental material does not provide a detailed description of the data processing pipeline as claimed, which raises concerns about reproducibility and the preparation of supervised training data.
- It is unclear how the network takes in echocardiograph images to estimate 3D heart shape, especially given that it requires contours of 2D echo planes (Figure 2).
- The paper claims early on that no paired data is needed, but it is not clear how this is enabled in the proposed method given the use of paired supervised data.
- The methods description does not clearly explain what "weak supervision" means or how it is applied, nor does it provide a clear understanding of the "contrastive" aspect of the formulation or learning process.
- There is no comparison with other relevant methods in the field, making it difficult to evaluate the proposed approach's performance and limitations.
- No statistical significance/equivalence tests are performed to support claims about reconstruction performance differences among different slices (Line 361).
- The authors rely heavily on Figure 2 to describe the method, but it is unclear how and why each subnetwork is structured in a particular way.
- Given the lack of clarity on data processing pipeline and network architecture, it would be challenging for readers to reproduce the results presented in the paper.

**Questions:**

- See weaknesses.
- Please provide a more detailed description of the data processing pipeline used to prepare your supervised training data.
- Can you provide a comprehensive statistical analysis of your results, including p-values, confidence intervals, and effect sizes?
- How do you define "weak supervision" in this context, and how does it differ from traditional supervised learning approaches? Please provide specific examples or illustrations to clarify this concept.
- Can you elaborate on the contrastive learning aspect of your method? Specifically, what is being contrasted (e.g., positive vs negative pairs), and how do you define these pairings? Additionally, how sensitive is the training process to batch size settings?
- How does the network estimate the 3D heart model given an echocardiographic image rather than contours? Please provide a detailed explanation of the architectural design and any key components or features that enable this capability.

---

### Official Review · Reviewer_2hLP · 2024-11-04

**Soundness:** 3
**Presentation:** 2
**Contribution:** 3
**Rating:** 5
**Confidence:** 3

**Summary:**

This paper proposes a 2D-to-3D inference approach to generate 3D heart geometry from 2D echocardiography. It uses multi-structure reconstruction loss, an overall reconstruction loss, and contrastive loss to enhance precision.

**Strengths:**

1. The problem has clinical significance.
2. Global heart structure and different views and parts are considered in the model design and evaluated separately.
3. The experiment includes quantitative and qualitative evaluation and ablation studies.

**Weaknesses:**

1. The method is not very clearly illustrated. Figure 2 is essential, but there is no caption for it. How are these decoders designed? What is the difference between the view completion and view reconstruction decoders? It seems that they can be merged into one decoder.
2. The paper proposed to solve the shape completion problem. But shape completion is a large collection of problems and need to be better defined in this paper. Shape completion seems to mean inferring 3D geometry from 2D planes, but what if some 2D views are missing? Does this model still work?
3. The figures need to be improved. For example, in Figure 5, curve plots (for showing the trends) are improper for comparing different heart components here. Bar plots should be better.
4. The evaluation metrics should have been better designed and illustrated. Are the geometric distances, e.g., the chamfer distances, calculated on normalized point clouds? However, the actual distances in mm are also desired. And how big is the heart, e.g., in mm? And how are the distances compared to the actual size of the heart? How applicable is this approach to real scenarios?
5. For localization, why the threshold is 2mm?

**Questions:**

See weaknesses.

---

### Note · Authors · 2024-12-02

**Comment:**

We are satisfied with obtaining many valuable opinions and withdraw the manuscript with gratitude.

**Withdrawal Confirmation:**

I have read and agree with the venue's withdrawal policy on behalf of myself and my co-authors.